# Revertant Mosaicism in Epidermolysis Bullosa

**DOI:** 10.3390/biomedicines10010114

**Published:** 2022-01-06

**Authors:** Cameron Meyer-Mueller, Mark J. Osborn, Jakub Tolar, Christina Boull, Christen L. Ebens

**Affiliations:** 1Medical School, University of Minnesota, Minneapolis, MN 55455, USA; meye3216@umn.edu; 2Department of Pediatrics, Division of Blood and Marrow Transplantation & Cellular Therapy, University of Minnesota, Minneapolis, MN 55455, USA; osbor026@umn.edu (M.J.O.); tolar003@umn.edu (J.T.); 3Stem Cell Institute, University of Minnesota, Minneapolis, MN 55455, USA; 4Department of Dermatology, Division of Pediatric Dermatology, University of Minnesota, Minneapolis, MN 55455, USA; oehr0005@umn.edu

**Keywords:** epidermolysis bullosa, revertant mosaicism, cellular therapy, gene therapy, autograft, loss of heterozygosity

## Abstract

Epidermolysis bullosa (EB) is a group of genetic blistering diseases characterized by mechanically fragile skin and mucocutaneous involvement. Historically, disease management has focused on supportive care. The development of new genetic, cellular, and recombinant protein therapies has shown promise, and this review summarizes a unique gene and cell therapy phenomenon termed revertant mosaicism (RM). RM is the spontaneous correction of a disease-causing mutation. It has been reported in most EB subtypes, some with relatively high frequency, and has been observed in both keratinocytes and fibroblasts. RM manifests as identifiable patches of unaffected, blister-resistant skin and can occur through a variety of molecular mechanisms, including true back mutation, intragenic crossover, mitotic gene conversion, and second-site mutation. RM cells represent a powerful autologous platform for therapy, and leveraging RM cells as a therapeutic substrate may avoid the inherent mutational risks of gene therapy/editing. However, further examination of the genomic integrity and long-term functionality of RM-derived cells, as well in vivo testing of systemic therapies with RM cells, is required to realize the full therapeutic promise of naturally occurring RM in EB.

## 1. Introduction

Epidermolysis bullosa (EB) is a group of blistering diseases characterized by mechanically fragile skin and mucocutaneous ulcerations both externally (skin and ocular) and internally (oral and gastrointestinal). The four major subtypes of EB are named for clinical features in combination with the anatomic level of blister formation in the skin (cleavage plane) and include EB simplex (EBS), junctional EB (JEB), dystrophic EB (DEB), and Kindler syndrome (Figure 1) [1]. Severity occurs across a spectrum within each subtype based on the location of the genomic coding alterations within the affected genes. Individuals are classified based on the extent of involvement as localized or generalized. EBS is the most common subtype and typically least severe, caused by aberrant protein formation in the epidermis. EBS most often results from a dominant-negative mutation in keratin 5 (*KRT5*) or 14 (*KRT14*), leading to basal intraepidermal cleavage. JEB is an autosomal recessive subtype affecting the dermal–epidermal junction proteins, including laminin-332 (three genes contributing: *LAMA3*, *LAMB3*, and *LAMC2*) and type XVII collagen (*COL17A1*) leading to intra-lamina lucida cleavage. Patients may display severe skin blistering and mucosal ulcerations with more severe disease typical when mutations affect laminin-332 as compared to type XVII collagen. DEB has both recessive and dominant inheritance patterns affecting proteins in the uppermost papillary dermis. DEB results from mutations in collagen VII (*COL7A1),* leading to sublamina densa cleavage. Recessive DEB (RDEB) is the most severe variant of EB, frequently associated with widespread blistering, mucosal and ocular disease, infection, fusion of the fingers and toes (pseudosyndactyly formation), and high risk of fatal cutaneous squamous cell carcinoma [2]. Finally, Kindler syndrome is an autosomal recessive disease affecting the protein kindlin-1 (*FERMT1),* leading to mixed cleavage planes. Inadequate protein connections between the skin layers cause poor skin integrity and leave the fragile skin vulnerable to cleavage, blister formation, and destruction from friction and mechanical trauma.

While EB has a wide variety of clinical manifestations, all forms significantly affect the quality of life for patients and caregivers [3]. Disease management has traditionally focused on supportive therapies, including wound care, pain control, and infection prevention. In the last decade, however, the development of genetic, cellular, and recombinant therapies has offered new treatment options for patients with EB, though these therapies have demonstrated variable success. 

Clinical trials of local wound therapies, including intradermal injection of allogeneic fibroblasts and mesenchymal stromal cells (MSCs) have shown only transient improvement in wound healing [4,5]. These localized therapies can be suboptimal as the migration of cells to other sites of involvement is limited. Clinical trials of systemic interventions aim to address both cutaneous and less accessible gastrointestinal mucosal manifestations of the disease. Of these, allogeneic hematopoietic cell transplant (alloHCT) serves as an important disease-modifying treatment modality but is reserved for those with the most severe phenotypes as it carries significant risks related to conditioning regimen toxicity and alloimmune reactions [6,7,8]. Despite this, alloHCT can provide a platform for establishing bidirectional immune tolerance, permitting ease of transfer of additional donor cells, including MSCs and skin grafts [8,9]. 

Gene therapy is a highly promising potential treatment for EB. Investigations examining the autologous transplant of gene-corrected keratinocytes with skin grafts are ongoing. However, genetic stability, as well as clinical safety, pose barriers to future implementation [10]. Gene therapy employs exogenous regulatory elements to drive gene expression of the therapeutic gene. However, genes such as *COL7A1* are prohibitively large for effective packaging in many viral vectors. Moreover, integrating vectors often show a promiscuous integration profile that can represent a safety risk. An approach that maximizes safety and maintains control of target gens under the endogenous regulatory elements is gene editing [11]. Because ex vivo engineering and expansion of gene-corrected cells can be challenging, autologous therapy with revertant mosaicism cells represents a “natural gene therapy” strategy.

## 2. Discovery of Revertant Mosaicism

RM refers to the spontaneous correction of a disease-causing mutation. The correction occurs most often during mitosis and results in at least two genetically heterogeneous cell populations. RM was first discovered in Lesch–Nyhan syndrome in 1988 [12] and was later identified in other inherited diseases, including EB [13,14]. RM was originally reported in EB in 1997 in a *COL17A1* JEB patient [13] who demonstrated clinically unaffected patches of skin on the hands and upper arms. Following PCR amplification, haplotype, and direct sequencing, researchers concluded that *COL17A1* gene conversion—nonreciprocal transfer at a specific gene locus where part of an allele was replaced by the homologous segment from another allele—was the most likely mechanism. RM has since been documented in EBS, JEB generalized intermediate, RDEB, DDEB, and Kindler syndrome, implicating *COL17A1*, *KRT14*, *LAMB3*, *COL7A1*, and *FERMT1* genes [15,16,17,18,19,20,21,22,23,24,25,26,27,28] (Figure 2, Table 1), and the topic of prior reviews [19,29,30]. Clinically, patches of revertant mosaic keratinocytes in *COL17A1* are identifiable as relatively hyperpigmented patches of skin compared to the surrounding affected epidermis [31]. 

Similar hyperpigmentation was also observed in *LAMB3* JEB but not RDEB or DDEB. RM skin patches do not show signs of blistering after mechanical stimulation (“complete” RM). However, “partial” RM has been described in cases where genotypic changes are consistent with RM, without phenotypic improvement of the associated skin [15,32].

RM in EB can occur through a variety of molecular mechanisms, including true back mutation, intragenic crossover, mitotic gene conversion, and second-site mutation (Figure 3). RM from alteration of mRNA via the splicing process [32], an independent frameshift nullifying dominant-negative allele [16], splice site mutation restoring a frameshift [17], indel, large base-pair substitution [31], and nucleotide substitution [33] have also been observed. Cases exist of multiple distinct RM events occurring in an individual patient, and the frequency of RM is so high in JEB generalized intermediate *COL17A1* patients (10 of 10 clinically, with 6 of 10 genetically confirmed in one cohort) it has been hypothesized that the majority if not all patients have genomic evidence of RM [31]. Previously, RM had been limited to keratinocytes. However, in 2019, our group reported the first evidence of RM in the fibroblasts of patients with EB [34].

## 3. Characterization of RM in EB

Laser capture microdissection was one of the first methods researchers used to isolate subpopulations of revertant keratinocytes overlying areas of basement membrane that positively stained for functional collagen [15]. This eliminated the need for culturing keratinocytes and eliminated possible selection bias from in vitro growth. Sequencing of PCR amplification products allowed broad categorization of the RM mechanism. However, RM is often caused by intragenic crossover, and the ability to compare single nucleotide polymorphisms (SNPs) between samples is often complicated by the size of genes causative of EB. In many cases, the exact mechanism of RM is unknown for precisely this reason, and researchers are unable to distinguish with certainty between back mutation and mitotic recombination. Advancements in sequencing have allowed more detailed analyses and methodologies to date include targeted RNA expression (TREx) assays and long-read approaches such as single-molecule real-time (SMRT) sequencing (PacBio) and nanopore RNA sequencing (Oxford Nanopore) [35]. These modern strategies permit better quantification of differential allele or SNP expression. 

## 4. Therapeutic Applications of RM in EB

RM restores gene expression while avoiding the potentially confounding safety factors related to gene therapy and editing, such as random vector integration or off-target effects. As such, RM cells represent a powerful autologous platform for therapy. Grafting is its most immediate application, initially attempted with autologous revertant keratinocytes from a patient with *COL17A1* JEB in 2009 [36]. RM keratinocytes were harvested by adhesive stripping and transferred to a wounded site. The attachment technique was successful, yet only 3% of graft keratinocytes were revertant compared to 30% in vitro, and the graft demonstrated subsequent blistering [36]. Subsequently, the first successful split-thickness biopsy graft of revertant cells of a *LAMB3* JEB patient delivered to wounded skin showed no blistering at the graft or donor sites at 18 months follow-up. Evaluation of both donor and graft sites confirmed normal laminin-332 by immunofluorescence [37]. Suction grafts utilizing heat and negative pressure to separate epidermis from dermis or punch grafts have additionally been utilized for the transfer of RM keratinocytes. While smaller grafts are easier to harvest, the acceptor sites of such grafts are equally small, and improved techniques for in vitro expansion are mandated. One approach is to generate pluripotent stem cells (iPSCs) from RM cells, as has been accomplished in RDEB and JEB [25]. IPSC represents a putative limitless population of cells, and their autologous nature mitigates the risk of immune rejection intrinsic to allogeneic grafts. RM keratinocytes derived from the iPSCs were made into a 3D skin equivalent with epidermis-like layers with the expression of functional collagen VII. iPSCs generated from revertant keratinocytes in a JEB patient were used to create in vitro 3D skin equivalents and reconstituted human skin in vivo in mice [38]. While the replication properties and broad differentiation potential represent a theoretically limitless population of cells, questions remain regarding the long-term viability and functionality of the iPSC-derived cells in vivo [39]. 

Long-term viability has been documented in the use of cultured epidermal autografts (CEAs) using RM keratinocytes. This technique was originally described and employed for burn patients and has been applied to RDEB patients using their own revertant cells. The graft site remained epithelized for 16 years and continued to show evidence of RM at both the donor and acceptor graft sites [40]. Based on these results, three patients with RDEB were given CEAs [41]. All ulcers showed improvement within two weeks, and two of the patients showed persistent epithelization for at least 76 weeks. Interestingly, the EB quality of life scores did not improve throughout the trial, possibly because grafted areas represented a relatively small portion of the total affected areas for patients. 

## 5. Future Steps to Expand Therapeutic Translation of RM

IPSC technology is promising, however, several limitations must be overcome before patients can widely benefit. Questions surrounding genomic integrity and long-term functionality of iPSC-derived cells in vivo warrant additional investigation. IPSC reprogramming methods are prone to concerning genetic alterations (single nucleotide variations, copy number variations) that could result in reduced differentiation capacity or malignancy [42]. Skin grafting, while an appropriate treatment for some EB patients, does not address the oral or intestinal mucosal wounds associated with certain EB subtypes, nor is it necessarily a feasible option for patients with widespread blistering. In such cases, systemic therapies such as alloHCT may be more advantageous. There is evidence that reduced-intensity conditioning alloHCT has led to systemic improvements, including decreased adverse effects and faster recovery [8,43,44]. Culturing autologous revertant cells for transplant would eliminate the necessity for aggressive immunosuppression regimens, but in vivo testing is lacking.

## 6. Conclusions

RM is an exciting natural phenomenon with expanding therapeutic applications, promising for both systemic and cutaneous treatments. We still know relatively little, however, regarding the nature of its mechanism and its ultimate role in EB treatment. Are these corrective mutations occurring randomly? A mathematical model developed to assess the probability of single nucleotide reversions in EB was found to underestimate the incidence of clinically observed RM in EB [45]. The authors hypothesized this could be explained by a “late-but-fitter revertant cell” theory, where revertant cells arise later in development but with a selective growth advantage leading to patches of unaffected skin. It was also hypothesized that the highly repetitive sequence in *COL7A1* and open chromatin structure in skin development could increase the likelihood of slipped DNA strand mispairing and other potential RM mechanisms [25]. Some research finds evidence of preferred mechanisms of RM [24], while others have found many different correction mechanisms for the same original mutation [17,31]. 

Additionally, it is difficult to pinpoint the chronology of RM during embryologic development. It is hypothesized that bilateral distribution of the same RM mutation suggests a timepoint before the second week of development prior to lateralization of the embryo [13]. There are often multiple genetically unique RMs in a single patient, and it is impossible to timestamp each RM confidently. The clinical progression of RM also varies, with some patients having patches of unaffected skin for as long as they can remember, while others have noticed newer expansion. Currently, there is no explanation for these differences. As most of the documented cases of RM are in adults, it would be valuable to observe RM in younger patients and document the progression of RM starting in infancy.

RM has been described in other genetic syndromes, especially those with increased cellular proliferation. Their varied mechanisms of RM may suggest that the physiology of RM is at least in part dependent on the disease process itself. Ichthyosis with confetti (IWC) is a genodermatosis characterized by hundreds to thousands of RM macules and patches that expand with age [30]. Interestingly, RM in IWC was found to occur solely from mitotic recombination, in contrast to RM in EB [30]. Wiskott–Aldrich syndrome and Fanconi anemia are hematopoietic system diseases defined by genomic instability, which may help to explain why individual patients with Wiskott–Aldrich syndrome have shown multiple distinct RM mechanisms [17,31]. Particularly in hematopoietic diseases, RM has shown disadvantages, including milder symptoms that mask diagnosis and delay treatment, precipitation of myelodysplastic syndrome, and immune system complications [46]. No disadvantages have been described in EB patients with RM, though the possibility of an RM resulting in additional undesirable genetic events exists.

Questions about the long-term durability, safety, and efficacy of RM-derived treatments must be addressed before the widespread use of RM in EB treatment. However, the discovery of RM in EB has stimulated a surge of exploration into its therapeutic potential (Figure 4) in both allogeneic tissues and for exogenous gene therapy. As the list of RM’s promising applications continues to expand, it seems likely that this “natural gene therapy” will find a significant place in the future treatment of EB.

## Figures and Tables

**Figure 1 biomedicines-10-00114-f001:**
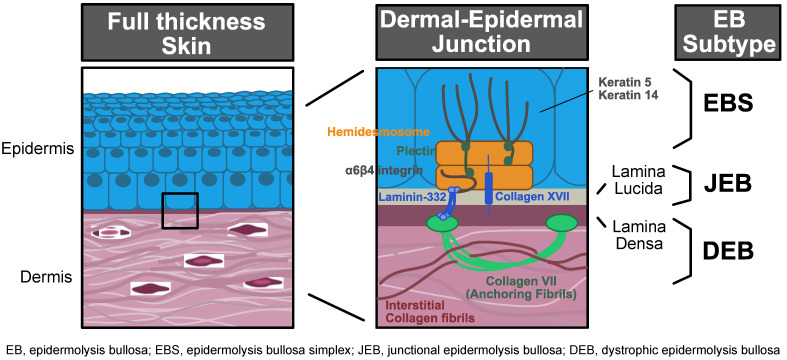
EB subtype schematic.

**Figure 2 biomedicines-10-00114-f002:**
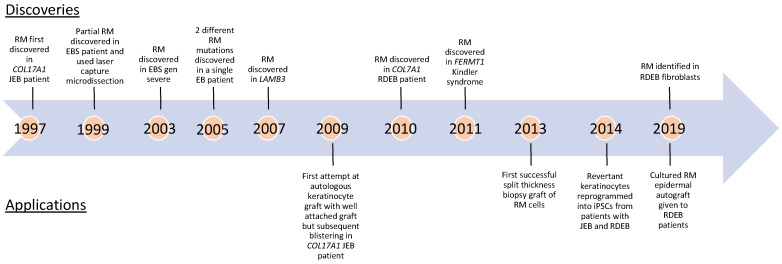
Timeline of discoveries in EB revertant mosaicism and clinical interventions.

**Figure 3 biomedicines-10-00114-f003:**
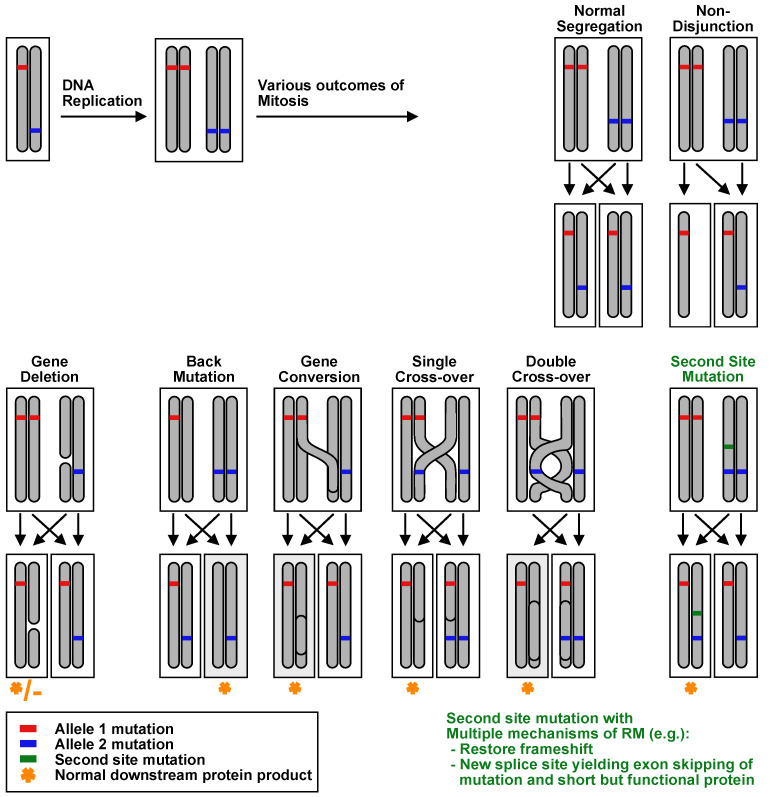
Mechanisms of revertant mosaicism in a compound heterozygous disease.

**Figure 4 biomedicines-10-00114-f004:**
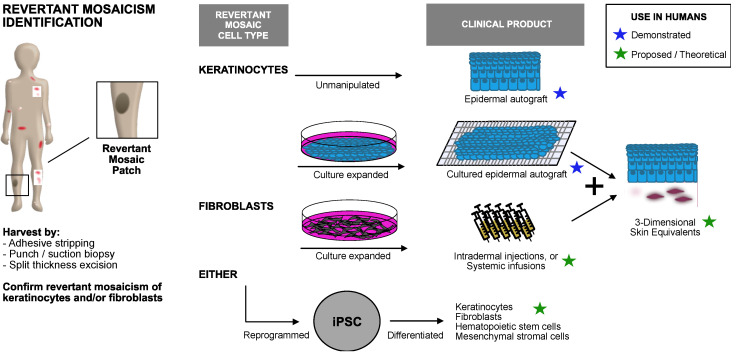
Clinical applications of revertant mosaic skin populations in EB.

**Table 1 biomedicines-10-00114-t001:** Published cases of revertant mosaicism in EB keratinocytes.

Age at Diagnosis of RM, Sex	Disease Causing Alleles ^1^	Partial or Complete Reversion	Reversion Mechanism	Size, Location, and Stability of Revertant Patches	Reference(s)
**JEB generalized intermediate (AR, *COL17A1* mutations)**					
28 y/o F	c.1601delA c.3676C > T (p.Arg1226X)	Complete Complete	Gene conversions (*n* = 3; upper extremities) Second site mutation, c. 3677G > C (left ankle)	Left upper arm, both lower arms, hands, left ankle, right lower leg, back and scalp (10% BSA). Some areas static, others with slow expansion.	Jonkman et al., 1997 [13]
					Pasmooij et al., 2005 [17] (Pt 2)
					Jonkman et al., 2009 [21] (Pt 3)
					Pasmooij et al., 2012 [31] (026-01)
56 y/o F	c.4003delTC (homozygous)	Partial	Second site mutation restoring reading frame lost with original two nucleotide deletion: c.4080insGG	N/A	Darling et al., 1999 [15]
75 y/o M	c.4320insC c.3676C > T (p.Arg1226X)	Complete Partial	Second site mutation in a splice site restoring frameshift: c.4358-1G > A (finger) Back mutation c.3676T > C or gene conversion (arm)	Right middle finger (2 cm^2^) and arm. Static.	Pasmooij et al., 2005 [17] (Pt 1)
					Jonkman et al., 2009 [21] (Pt 8)
					Pasmooij et al., 2012 [31] (093-01)
46 y/o M	c.2237delG (p.Gly746AlafsX53) c.3676C > T (p.Arg1226X)	Complete	Second site mutation c.2263 + 2T > C	Bilat hands and lower arms, left upper arm, forehead, face	Jonkman et al., 2009 [21] (Pt 4)
					Pasmooij et al., 2012 [31] (035-02)
48 y/o M	c.2237delG (p.Gly746AlafsX53) c.3676C > T (p.Arg1226X)	Complete	Second site mutations: c.2237insG (or gene conversion), c.2263 + 2T > C, Del14(2259–2263 + 9)	N/D	Jonkman et al., 2009 [21] (Pt 5)
46 y/o M	c.2237delG (p.Gly746AlafsX53) (homozygous)	Complete	Second site mutations resulting in exon 30 skipping (location of disease-causing mutation): c.2238C > T, c.2227 + 153_2336-318del (large deletion, additionally skipping exon 31)	Wrist	Jonkman et al., 2009 [21] (Pt 13)
					Pasmooij et al., 2012 [31] (208-01)
59 y/o F	c.2237delG (p.Gly746AlafsX53) (homozygous)	Complete	Second site mutation resulting in skipping of exon 30 (location of disease-causing mutation): c.2263_2T > C	Hands, arms, and back	Pasmooij et al., 2012 [31] (025-01)
48 y/o M	c.2237delG (p.Gly746AlafsX53) c.3676C > T (p.Arg1226X)	Complete	Second site mutations resulting in exon 30 skipping (location of disease-causing mutation): c.2228-101_2263 + 70delins15 (indel), c.2259_2263 + 9del, c.2263 + 2T > C	Bilat knees and hands, patches on right upper leg	Pasmooij et al., 2012 [31] (035-01)
21 y/o F	c.3487G > T (p.Glu1163Ter) c.1490_1491delinsT (p.Ala497Valfs*23)	Complete	Second site mutation in splice site resulting in exon 49 skipping (location of disease-causing mutation): c.3419-1G > T	Right lower arm	Kowalewski et al., 2016 [22]
**EBS (AR, *KRT14*)**					
67 y/o F	c.526-2A > C (homozygous)	Partial	Second site mutations disrupting splice site acceptor created by original mutation: c.528T > G, c.529del6 (identified not in DNA but in mRNA)	N/A	Schuilenga-Hut et al., 2002 [32]
**EBS (AD, *KRT14*)**					
Early 20′s F	c.373C > T (p.Arg125Cys)	Complete	Second site mutation creating a premature termination codon nullifying the downstream dominant negative allele: c.242insG	Trunk blistering resolved over teen years. Extension with time.	Smith et al., 2004 [16]
**JEB generalized intermediate (AR, *LAMB3*)**					
46 y/o M	c.628G > A (p.Glu210Lys) c.1903C > T (p.Arg635X)	Complete	Multiple second site mutations: c.628 + 42G > A, c.596G > C	Left lower leg. Extension with time.	Pasmooij et al., 2007 [20] (078-01)
63 y/o M	c.628G > A (homozygous)	Complete	Multiple second site mutations: c.565-3T > C, c.619A > C, p.Lys207Gln, c.629-1G > A	Arm, shoulder, chest. Static.	Pasmooij et al., 2007 [20] (029-01)
**RDEB generalized severe (AR, *COL7A1*)**					
41 y/o M	c.1732C > T (p.Arg578X) c.7786delG (p.Gly2593fsX4)	Complete	Intragenic crossover somewhere between the two mutations yielding one normal allele and one with c.7786delG mutation (Left wrist)	Left wrist, right shin (up to 8 × 5 cm). Static.	Almaani et al., 2010 [18]
42 y/o F	c.6527insC (homozygous)	Complete	Second site mutation correcting the reading frame of the original mutation: c.6528delT	Left forearm (8 × 4.5 cm). Static.	Pasmooij et al., 2010 [23]
21 y/o M	c.6508C > T (p.Gln2170X) (homozygous)	Complete	Second site mutation restoring nonsense codon created by original mutation: p.6510G > T (p.Gln2170Tyr)	Patch on right lateral neck (2.5 × 3 cm). Static.	Van den Akker et al., 2012 [33] (EB024)
63 y/o M	c.425A > G c.8206G > A (p.Glu2736Lys)	Complete	Mitotic recombination resulting in loss of original c.425A > G mutation (which had caused altered splicing and a premature termination codon), but noted LOH of neighboring SNP c.2945A > G (p.Pro939Pro) thus deemed not to result from a back mutation	Left lower leg (two 3 × 3 cm patches). Static.	Kiritsi et al., 2014 [24] #2
21 y/o	c.2142A > G c.6527dupC (p.Gly2177Trpfs*113)	Complete	Second site mutation restoring splice site affected by original c.2144A > G mutation	Dorsum of right hand (7 × 3 cm)	Kiritsi et al., 2014 [24] #3
22 y/o	c.884delG c.6527dupC (p.Gly2177Trpfs*113)	Complete	Back mutation or mitotic recombination (unable to further differentiate) resulting in loss of original c.884delG mutation	Right lower arm (7 × 4 cm). Noted at 14 years of age.	Kiritsi et al., 2014 [24] #4
37 y/o	c.425A > G (homozygous)	Complete	Second site mutation restoring normal splice site caused by original c.425A > G mutation: c.426 + 3G > A	Lateral lower leg (4 × 4 cm). Static.	Kiritsi et al., 2014 [24] #5
17 y/o	c.425A > G c.1837C > T (p.Arg613*)	Complete	Mitotic recombination suggested as both original mutations detected and no additional mutations detected (presume recombination event placed original mutations on 1 allele with other allele without mutations)	Right hand	Kiritsi et al., 2014 [24] #6
12 y/o	c.4894C > T (p.Arg1632*) c.6176A > G (p.Glu2059Gly)	Complete	Back mutation/mitotic recombination (unable to further differentiate) resulting in loss of original c.6176A > G mutation	Back (10 × 5 cm), Lateral right leg (5 × 3 cm). Static.	Kiritsi et al., 2014 [24] #7
Birth M	c.3840delC (p.Thr1280Thrfs*44) c.6751-2A > G (IVS85-sA > G)	Complete	Uncertain mechanism resulting in the retention of original c.3840delC mutation but skipping of exon 86 (3 outcomes: 6 bp skipped, 10 bp skipped and entire exon 86 skipped) downstream of c.6751-2A > G exon 85 acceptor splice site mutation	Pubic region (7 × 11 cm). Static.	Tolar et al., 2014 [25]
**Kindler syndrome (AR, *FERMT1*)**					
22 y/o M	N/D	Complete	N/D (Normal skin structure of RM patch on biopsy)	Dorsal feet, left palm (4 cm), neck. Static.	Al Aboud et al., 2003 [26]
Birth M	c.676dupC (p.Gln226fsX17) (homozygous)	Complete	Transcriptional slippage or RNA editing: Loss of extra cytosine in mRNA despite genomic DNA still containing the mutation	Right hand. Static.	Lai-Cheong et al., 2012 [27]
29 y/o M	c.456dupA (p.Asp153ArgfsX4) (homozygous)	Complete	Back mutation resulting in loss of the adenosine duplication and restoration of the reading frame on a single allele	Innumerable lesions of the entire integument (several mm^2^ to 15 cm^2^)	Kiritsi et al., 2012 [28] (P1)
24 y/o F	c.676dupC (p.Gln226ProfsX17) (homozygous)	Complete	Back mutation resulting in loss of the cytosine duplication and restoration of the reading frame on a single allele	Hands, lower legs (0.5 cm^2^ to 2 cm^2^)	Kiritsi et al., 2012 [28] (P2)
17 y/o F	c.676dupC (p.Gln226ProfsX17) (homozygous)	Complete	N/D, authors presume heterozygous back mutation restoring reading frame	Hands (0.5 cm^2^ to 3 cm^2^)	Kiritsi et al., 2012 [28] (P3)
21 y/o F	c.676dupC (p.Gln226ProfsX17) (homozygous)	Complete	N/D, authors presume heterozygous back mutation restoring reading frame	Hands, neck, legs (0.5 cm^2^ to 3 cm^2^)	Kiritsi et al., 2012 [28] (P4)
11 y/o F	c.676dupC (p.Gln226ProfsX17) (homozygous)	Complete	N/D, authors presume heterozygous back mutation restoring reading frame	Hands, lower legs (0.5 cm^2^ to 2 cm^2^)	Kiritsi et al., 2012 [28] (P5)
9 y/o M	c.676dupC (p.Gln226ProfsX17) c.1677G > A (p.Trp559X)	Complete	N/D, authors presume heterozygous back mutation restoring reading frame	Hands, arms, legs (several mm^2^ to 1 cm^2^)	Kiritsi et al., 2012 [28] (P6)
**DDEB (AD, *COL7A1*)**					
23 y/o	c.6127G > A (p.Gly2043Arg)	Complete	Back mutation/mitotic recombination restoring normal sequence: c.6127C > A	Right arm (3 × 3 cm)	Kiritsi et al., 2014 [24] #1

^1^ Gene mutation nomenclature: modern reporting counts nucleotides from the start codon, cases reported from Jonkman et al., 1997, Pasmooij et al., 2005, and Schuilenga-Hut et al., 2002 updated to reflect such practice. RM, revertant mosaicism; JEB, junctional epidermolysis bullosa; AR, autosomal recessive; y/o, years old; F, female; BSA, body surface area; pt, patient; N/A, not applicable; M, male; N/D, not disclosed; EBS, epidermolysis bullosa simplex; AD, autosomal dominant; RDEB, recessive dystrophic epidermolysis bullosa; LOH, loss of heterozygosity; SNP, single nucleotide polymorphism; DDEB, dominant dystrophic epidermolysis bullosa. #1: This patient is listed as “#1” in the referenced publication.

## Data Availability

Data sharing is not applicable to this article as no new data were created or analyzed.

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
