# Peer review of "Revertant Mosaicism in Epidermolysis Bullosa"

_biomedicines, 2022, doi:10.3390/biomedicines10010114_

Round 1

Reviewer 1 Report

This review paper is well-written, concise and well-constructed on revertant mosaicism in erpidermolysis bullosa. The reviewer enjoyed the paper, and it will be of interest for clinical dermatologists and researchers.

Minor point; 

Page 1, Line 43: Laminin-332 does not refer to LAMB3 but LAMA3, LAMB3 and LAMC2.

Author Response

We thank the reviewer for the kind support of our review and agree with your minor point. Page 1 Line 43 has been edited to reflect all 3 genes contributing to subunits of the glycoprotein complex laminin-332.

Reviewer 2 Report

The article is a narrative review about Revertant mosaicism in epidermolysis bullosa. Although the article is interesting and well written, there are various similar articles in medical literature, so I do not see any big novelty to justify publication in a highly impacted journal; 

Author Response

We appreciate the candid feedback from the reviewer and acknowledge revertant mosaicism in epidermolysis bullosa has been previously summarized in the literature. This manuscript was an invited review with the topic dictated by the special edition editor. We provide a comprehensive, updated review of all published cases, including the new discovery of revertant mosaicism in epidermolysis bullosa fibroblasts, describe prior efforts to leverage this natural gene therapy for therapeutic benefit, and propose potential future avenues. Further, we prompted readers to consider existing questions regarding (1) the timing of revertant mosaic events, (2) inconsistencies in outgrowth, stability, or contraction of revertant mosaic patches with time and whether selective advantage plays a role, (3) the interaction of the epidermolysis bullosa disease process and occurrence of revertant mosaicism, and (4) need for further study for safe and successful translation of revertant mosaicism from an observation into the basis for a therapeutic. The novelty we provide is reviewing revertant mosaicism from a new perspective, namely the perspective of translational researchers, highlighting what gaps exist in current knowledge of revertant mosaicism before potential new therapeutic approaches to epidermolysis bullosa can be derived from revertant mosaicism. 

Reviewer 3 Report

Sir, 

I have recently reviewed the manuscript "Revertant mosaicism in epidermolysis bullosa" submitted by Cameron Meyer-Mueller and co-workers in Biomedicines (biomedicines-1518288). In this comprehensive review, the authors aimed to summarise the current knowledge on revertant mosaicism and present the example of a rare genetic disorder - epidermolysis bullosa. I am very glad to see that such effort is dedicated to this interesting genodermatosis.

The structure of the review is generally well balanced. The text is comprehensive and can attract the interest of a novice reader as well as of expert clinician. 

However, there are certain necessary corrections. Relevant to line 62, Reference 4 ... the original work mentioned intraDERMAL injection, not intraepidermal. Such an attempt would not make much sense. 

Further, the authors listed several possible mechanisms of RM (lines 110-119). Moreover, the authors also introduce the concept of alloHCT for EB (dystrophic type) treatment.  This inspires me to make a simple (and maybe naive) request. I  believe that it would be beneficial to consider and clearly discuss also an alternative scenario.  Historically, many papers have reported that "transdifferentiation" of circulating hematopoietic stem cells is an extremely rare event. I believe it is possible. How can the authors exclude that this is not another possible mechanism in spontaneous RM observed in EB? This would be a non-genetic mechanism, of course, as the transdifferentiating cell would not bear the mutation.  The special issue theme is "Somatic Mosaicism in Skin Disorders". But is it indispensable to see EB as a disorder affecting all three germ layers? In some milder cases of EBS, the blistering can be relatively focal (I have seen patients of this type). Then, we might understand the disease as mosaicism occurring on the background of normal epidermis of ectodermal origin.  In severe JEB, two germ layers would be presumably affected.  This (de novo) mutation would appear later during embryogenesis. If so, other germ layer/s would not bear the mutation, and this could be a reservoir for the "natural gene therapy" strategy (as mentioned by the authors  - line 81). 

I would be extremely grateful to the authors to respond to my request and consider preparing a brief explanatory paragraph on this issue that would be beneficial to me and also to others. 

Otherwise, I believe that this text is worthy of publication and represents a valuable contribution to current knowledge. 

Author Response

We thank the reviewer for your thoughtful comments. With regard to line 62, reference 4, you are correct that the injection site was "intradermal" not "intraepidermal." Thank you for your careful review and recognizing this error which has been corrected. Regarding mechanisms of revertant mosaicism and the concept of transdifferentiation, it is our understanding that in transdifferentiation, an HSC would differentiate into another cell type such as a keratinocyte. Certainly this is a mechanism proposed in the field of allogeneic hematopoietic stem cell transplantation. However, as EB is the result of a germline mutation and all cells in the body carry the same pathogenic mutation, transdifferentiation would still require a revertant mosaicism event to produce healthy skin cells. If there is another process by which this may occur, please provide additional clarifying feedback so we can appropriately address this question.

Round 2

Reviewer 2 Report

Given the fact that the paper was an invited review, and that overall is a well-written paper, it may be granted publication

Author Response

Thank-you for your time and efforts reviewing this manuscript.

Reviewer 3 Report

Sir, 

I have reviewed the manuscript and rebuttal letter provided by the authors. I am glad to see that they have followed the given instructions well. 

The topic which I suggested in my previous review is just a minor point. To clarify this issue more specifically, I would probably consider (a brief) discussion of somatic mosaicism as it was mentioned (RARELY - I agree) by e.g. following publications: 

From Marcel Jonkman´s group:  https://doi.org/10.1111/bjd.13336

and also in Shipman:  doi:10.1001/jamadermatol.2014.281. (present a nice comparison of lesional and nonlesional skin). 

As mentioned here, "Forward, or nonrevertant, mosaicism occurs during embryogenesis, when a mutation occurs in mitosis affecting only that subsequent cell line and not the other dividing cells of the embryo. The later it occurs during embryogenesis, the fewer cells will be affected."

I acknowledge that this is just a detail, but I believe this might be perfecting the publication in the special issue dedicated to mosaicism. 

However, I consistently and consciously regard this publication as a well written, concise and valuable review.  The point may be (or maybe not) reflected by the authors if they wish. No need to review this work by me anymore.  

Author Response

While we agree this is an important nuanced topic in the field of somatic mosaicism, we do not feel enough evidence exists to further describe the possible timing of the event leading to mosaicism. We do however, raise this as an ongoing area of interest and potential future investigation in the paragraph beginning on line 226. We have not edited the manuscript to incorporate further discussion.